# Early onset adult deafness in the Rhodesian Ridgeback dog is associated with an in-frame deletion in the *EPS8L2* gene

Takeshi Kawakami[1]*, Vandana Raghavan[1], Alison L. Ruhe[1,2], Meghan K. Jensen[1], Ausra Milano[1], Thomas C. Nelson[1], Adam R. Boyko[1,3]*

**1** Embark Veterinary, Inc., Boston, Massachusetts, United States of America, **2** ProjectDog, Davis, CA, United States of America, **3** Department of Biomedical Sciences, College of Veterinary Medicine, Cornell University, Ithaca, New York, United States of America

* adam@embarkvet.com (ARB); tkawakami@embarkvet.com (TK)

**Data Availability Statement:** All data necessary to replicate our results, including the array genotype

## Abstract

Domestic dogs exhibit diverse types of both congenital and non-congenital hearing losses. Rhodesian Ridgebacks can suffer from a progressive hearing loss in the early stage of their life, a condition known as early onset adult deafness (EOAD), where they lose their hearing ability within 1–2 years after birth. In order to investigate the genetic basis of this hereditary hearing disorder, we performed a genome-wide association study (GWAS) by using a sample of 23 affected and 162 control Rhodesian Ridgebacks. We identified a genomic region on canine chromosome 18 (CFA18) that is strongly associated with EOAD, and our subsequent targeted Sanger sequencing analysis identified a 12-bp inframe deletion in *EPS8L2* (CFA18:25,868,739–25,868,751 in the UMICH_Zoey_3.1/canFam5 reference genome build). Additional genotyping confirmed a strong association between the 12-bp deletion and EOAD, where all affected dogs were homozygous for the deletion, while none of the control dogs was a deletion homozygote. A segregation pattern of this deletion in a 2-generation nuclear family indicated an autosomal recessive mode of inheritance. Since *EPS8L2* plays a critical role in the maintenance and integrity of the inner ear hair cells in humans and other mammals, the inframe deletion found in this study represents a strong candidate causal mutation for EOAD in Rhodesian Ridgebacks. Genetic and clinical similarities between childhood deafness in humans and EOAD in Rhodesian Ridgebacks emphasizes the potential value of this dog breed in translational research in hereditary hearing disorders.

## Introduction

Hearing loss is the fourth leading cause of disability in humans, where about half of the population between the ages of 60 and 69 years experience some hearing loss [1, 2]. Age-related hearing loss, referred to as presbycusis, is a slow but progressive loss of hearing commonly occurring in the elderly and is a complex sensory disorder with a mixture of genetic and environmental components [3]. Congenital deafness, hearing loss present at birth, is another form

data, are freely available on Dryad with the provided DOI (doi:10.5061/dryad.pg4f4qrqt). Raw sequence reads of the whole genome sequencing analysis are available via the Short Read Archive (ncbi.nlm.nih.gov/sra; Bioproject number: PRJNA780814). Sanger sequence results are available in Genbank (GenBank accession: OL652584-OL652592).

**Funding:** This study was funded by Embark Veterinary, Inc. and the participants that provided DNA and phenotypic information via Embark's web-based platform. The funder only provided financial support in the form of salaries for all authors but did not have any additional role in the study design, data collection and analysis, decision to publish, or preparation of the manuscript.

**Competing interests:** I have read the journal's policy and the authors of this manuscript have the following competing interests: TK, VR, ALR, MKJ, AM, TCN, and ARB are employees of Embark Veterinary, a canine DNA testing company which offers commercial testing for the variant described in this study. This does not alter our adherence to PLOS ONE policies on sharing data and materials. ARB is co-founder and part owner of Embark.

of hearing disability which can have a substantial impact on the development of speech and learning skills [4, 5]. Although congenital deafness may be attributed to environmental factors, such as infections and maternal conditions, the majority of congenital cases can be attributed to genetic factors [6]. In addition to congenital deafness in newborns, hearing loss can take place neonatally or during early childhood. Similar to congenital deafness, the etiology of childhood hearing loss is diverse with a complex interaction of genetic and environmental components, but different etiological patterns, such as age of onset and progression of the hearing condition, appear to be correlated with specific genes [7]. Currently, more than 100 genes have been reported to be involved in non-syndromic hearing loss in humans [8], and these genes comprise a part of a genetic screening panel for early identification and management of congenital and early childhood hearing loss [7, 9]. Thus, continuous improvement of the genetic screening panel is of paramount importance for an accurate diagnosis of hearing loss. However, detailed interactions between genes and the relative contribution of non-genetic factors are unknown, and, as a result, evaluation of risk factors for this highly prevalent disability remains challenging.

Domestic dogs offer an excellent animal model system to explore the genetics of hearing loss because they exhibit a wide variety of hearing disorders, including congenital deafness, early and late onset of hearing loss, many of which are etiologically similar to those in humans. For instance, late or adulthood onset deafness has been reported in the Border Collie where the onset of hearing loss is 3–5 years old, but much later onset has also been reported [10, 11]. An analogous form of presbycusis in humans has also been reported in cross-sectional and longitudinal studies by using medium-sized mixed breed dogs with an onset around 8–10 years of age [12]. While the adult onset deafness in Border Collies may represent a hereditary sensorineural deafness with a potentially monogenic basis, the latter form of deafness could be due to a compound effect of noise exposure, aging-related neuronal cell degeneration in the cochlea, and potential genetic effect that may increase susceptibility to deafness [12, 13]. Many of the recognized forms of canine congenital deafness are pigmentation-related, such as deafness associated with the white spotting locus (S-locus) and the merle locus (M-locus), as these pigmentation genes play a crucial role in melanoblast migration and differentiation and the development of inner ear during embryogenesis [14, 15]. However, there are also known cases of pigmentation-unrelated congenital deafness in some breeds, including Doberman Pinschers [16–18] and Labrador Retrievers [14]. Independent identification of putative causal mutations in a gene *PTPRQ* in dogs and humans [19, 20] emphasizes the importance of dogs as a model system for studying naturally occurring hearing loss.

Rhodesian Ridgebacks exhibit a progressive postnatal deafness which may be observed as early as four months of age but more commonly observed within 1–2 years after birth [14]. This form of deafness, hereafter referred to as early onset adult deafness (EOAD), does not appear to be restricted to a specific bloodline because it has been identified in multiple Rhodesian Ridgeback populations, including North America, Europe, and Africa. A suspected mode of inheritance is autosomal recessive with one major locus on one of the autosomes, but this has yet to be experimentally confirmed. All affected dogs are visually indistinguishable from dogs with normal hearing by having fully pigmented coat, eyes, and noses, indicating that EOAD in Rhodesian Ridgebacks likely has a different genetic basis from pigmentation-related deafness. Since EOAD in Rhodesian Ridgebacks is unique by exhibiting much earlier onset than adult deafness in other breeds but is similar to some of the childhood deafness in humans, this condition offers a unique opportunity to identify novel genetic risk factors in canine deafness as well as to explore possible translational medical values for better understanding childhood hearing loss in humans.

Here we investigated genomic regions associated with EOAD in Rhodesian Ridgebacks by using a total of 185 dogs that were genotyped at 228,061 markers covering all 38 autosomes and chromosome X. Rhodesian Ridgeback owners contributed to this study by responding to an online survey about hearing ability in their dogs. Survey responses were used to identify genomic regions associated with EOAD by genome-wide association study (GWAS). Sequencing analysis of the associated region revealed a 12-bp deletion in *EPS8L2* as a strong candidate mutation for EOAD. As *ESP8L2* is expressed in cochlear hair cells and associated with recessive forms of childhood-onset deafness in humans, future studies of this form of deafness may provide important insights into deafness in humans and other animals.

## Materials and methods

### Samples

In order to recruit EOAD-affected and normal hearing dogs, we conducted an online survey in 2020 to private dog owners of Rhodesian Ridgebacks who were registered in the Embark customer database and agreed to participate in scientific research. The survey asked if 1) a dog is deaf (bilaterally or unilaterally) and, if yes, 2) suspected age of onset of deafness. In addition, EOAD-affected dogs were also recruited via ProjectDOG (https://www.projectdog.org/), where dog owners reported their dogs showing early development of hearing impairment with clinical signs of EOAD. Hearing loss was initially determined by owners to ensure that a dog likely had hearing capability at birth but subsequently exhibited hearing loss before reaching adulthood (less than 2 years old). In addition, the loss of hearing was clinically confirmed in all case dogs by Brainstem Auditory Evoked Response (BAER) test [21, 22]. BAER testing was coordinated by ProjectDog through individual clinics and in-home visits. Selected control dogs were also BAER tested to confirm normal hearing. Control dogs with normal hearing were recruited by the survey. All control dogs were at least five years old at the time of the study to ensure that they did not exhibit hearing loss. This dataset was used for GWAS to identify genomic regions and variants associated with EOAD (hereafter referred to as a discovery dataset).

Three additional EOAD-affected and 12 control Rhodesian Ridgebacks were recruited in 2021 after the initial discovery of the associated genomic region. These three cases were confirmed to be EOAD by observing early loss of hearing by the owners, followed by BAER testing, which was conducted at individual clinics. Two were full siblings (dog_1187 and dog_1188), and their sire (dog_1186) was one of 12 control dogs. The remaining EOAD-affected dog was unrelated to any of the dogs used in this study. These dogs were used as a validation dataset to confirm the association between a putative causal mutation and EOAD. All samples used for the initial discovery and follow-up validation are listed in S1 Table.

### Genotyping and genome-wide association

Buccal swabbing was conducted by dog owners to collect cheek cell samples for DNA extraction. DNA was extracted by Illumina, Inc. These DNA samples were genotyped using Embark's custom high-density SNP arrays (221,412 markers on 38 autosomes and 6,649 markers on the X chromosome). Genotyping rate calculation, pi_hat estimate, and marker filtering were performed by PLINK v1.9 [23].

A genomic region associated with EOAD in the Rhodesian Ridgeback was identified by using a multivariate linear mixed model implemented in GEMMA v.0.98 [24]. To minimize the confounding effects of shared ancestry between dogs, a kinship matrix was generated by GEMMA from the filtered genotype dataset and used as a random effect in the model. A Wald test was applied to detect significant association between markers and phenotype with a threshold of $P < 5.0 \times 10^{-8}$. To evaluate marginally significant associations identified in the

main GEMMA analysis, we repeated the analysis with the same number of case and control samples (10 and 107, respectively) out of the entire dataset including related individuals (23 case and 162 control dogs) by randomization with replacement for 100 times. We only considered regions showing significant association consistently across the 100 replications as a candidate EOAD-associated region in downstream analyses. To identify haplotypes associated with EOAD, we phased genotypes of markers on CFA18 of all genotyped Rhodesian Ridgebacks with Beagle v4.1 with default parameter settings [25].

## Sequencing analysis

Genomic DNA was extracted from one EOAD-affected dog and sequenced on an Illumina Genome Analyzer IIx to produce 100 bp paired-end reads. Raw sequence reads were mapped to the CanFam3.1 dog reference assembly by following the GATK best practice [26]. Briefly, adapter-marked sequence reads by MarkIlluminaAdapters were mapped to the reference genomes by using the BWA-MEM algorithm in BWA 0.7.17 [27], followed by duplicate marking by MarkDuplicates in Picard tools (http://sourceforge.net/projects/picard/). Base quality score was calibrated with BaseRecalibrator and ApplyBQSR in GATK version 4.1.9 [28]. Approximately 570 million reads were produced and 99.2% of the reads mapped to the dog reference genome. Sequence variants were called using HaplotypeCaller in GATK. Potential functional impacts of these variants were evaluated by using the Variant Effect Predictor (VEP) version 99.2 (http://www.ensembl.org/vep).

To identify large structural variations (SVs) in the EOAD-associated region on CFA18, we used Manta [29], which uses paired and split-read evidence for SV detection in mapped sequencing reads. In addition to the EOAD-affected Rhodesian Ridgeback, we downloaded whole-genome sequence data for 4 Rhodesian Ridgebacks and 37 dogs of the three additional breeds from the NCBI Sequence Read Archive (https://www.ncbi.nlm.nih.gov/sra) (S2 Table). We selected Pembroke Welsh Corgis, German Shepherd Dogs, and Labrador Retrievers as additional reference breeds because 1) a core EOAD-associated haplotype identified in Rhodesian Ridgebacks was identified in these breeds, and 2) these breeds are not known to exhibit EOAD. Sequence reads of these samples were mapped to the CanFam3.1 reference genome by using the method described above.

To characterize the genetic variations in a candidate gene *EPS8L2*, we designed 50 PCR primers to amplify regions in and around this gene (S3 Table). Amplicons were purified by Monarch PCR and DNA purification kit and were sequenced by the Sanger method at the Cornell Genomics Facility.

## Ethics statement

Participating dogs were part of the Embark Veterinary, Inc., customer base. Owners provided informed consent to use their dogs' data in scientific research. Non-invasive methods for genotype and phenotype collection were used, and dogs were never handled directly by researchers. Owners were given the opportunity to opt-out of the study at any time during data collection. All published data have been de-identified of all personal information as detailed in Embark's privacy policy (embarkvet.com/privacy-policy/).

## Results

### Early-Onset of Adult Deafness (EOAD) in Rhodesian Ridgeback

We identified 23 EOAD-affected Rhodesian Ridgebacks (11 males and 12 females) that were confirmed deaf by BAER testing; these dogs lost their hearing approximately from 6 months to

24 months after birth. All of the dogs were bilaterally deaf. We also identified 162 adult dogs (five years or older) with normal hearing as reported by their owners in the online surveys. Among these, six dogs were BAER tested, confirming their normal hearing. Some of the 23 EOAD-affected dogs were closely related with each other with the average relatedness being higher than the control group (pi-hat 0.226 vs 0.176). Our EOAD case dogs included one 2-generation family with seven members where two offspring both had EOAD, while their parents and other relatives exhibited normal hearing (S1 Fig). These samples were used in the downstream analyses to identify EOAD-associated regions by GWAS and screen candidate variants by haplotype analysis.

## A novel association on chromosome 18 with EOAD

Closely related dogs, defined as pairs of dogs with pi_hat > 0.45, were removed from the dataset used for the initial discovery of the associated genomic region by GWAS; 13 EOAD case dogs and 53 control dogs were removed. After removing markers with more than 3% missingness across the remaining dogs, we used 192,249 markers for GWAS, for which the genotyping rate was 99.8% in 117 dogs (10 case dogs and 107 control dogs) (S1 Table).

Our GWAS of 10 unrelated EOAD-affected dogs and 107 control dogs identified one highly significant region on CFA18 and two other marginally significant regions on CFA24 and CFA26 (Figs 1 and S2). The most significant marker was at the position 25,448,444 on CFA18, an intronic region of SEPTIN14 gene ($P = 5.8 \times 10^{-35}$). All affected dogs were GG homozygotes at this marker, whereas none of the control dogs carried the GG genotype (94 and 13 control dogs were AA homozygotes and AG heterozygotes, respectively) (S1 Table). The other two significant markers were at CFA24:42,714,739 (intergenic region, $P = 8.8 \times 10^{-8}$) and CFA26:23,554,170 (intronic region of MTMR3, $P = 2.0 \times 10^{-9}$). These two markers were not consistently detected as significantly associated regions in replicated runs, where we repeated the GWAS analysis 100

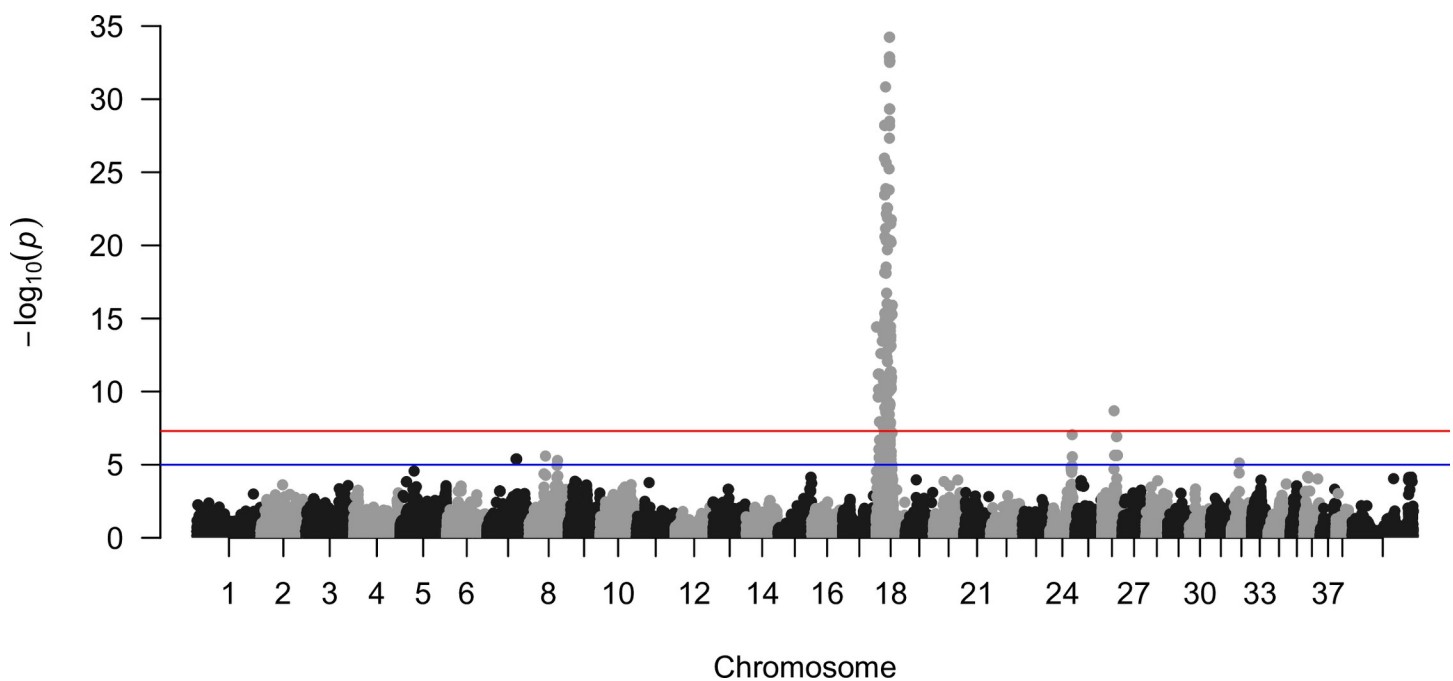

**Fig 1. A Manhattan plot of association with Early Onset Adult Deafness (EOAD) in Rhodesian Ridgebacks.** Red and blue horizontal lines are significant ($P < 5 \times 10^{-8}$) and suggestive ($P < 1 \times 10^{-5}$) associations, respectively.

times by randomly selecting 10 case and 107 control dogs from the entire dataset of 23 case and 162 control dogs (S4 Table). CFA24:42,714,739 was not significantly associated with EOAD in any of the replicated runs, and CFA26:23,554,170 was significantly associated with EOAD in 14 of the 100 replicated runs. In contrast, the CFA18 region was consistently significantly associated with EOAD across these replicated analyses, and thus, we only consider the region on CFA18 as a candidate region of EOAD in the following analysis.

Out of all EOAD affected dogs including closely related individuals (n = 23), 19 dogs were homozygous for a common haplotype in the associated region on CFA18, defined as a 1-Mb region from the position 24,891,130 to 25,897,356 where all markers showed Bonferroni corrected P values $< 1.0 \times 10^{-8}$ in the GWAS (Fig 2). The remaining four EOAD-affected dogs had one or two copies of closely related haplotypes, which were different from the common haplotype by one to five SNPs out of 140 markers. By including these closely related haplotypes as EOAD-associated haplotypes, all of the EOAD affected dogs were homozygous for an EOAD-associated haplotype. In the 162 control dogs, the most common EOAD-associated haplotype was found in 20 dogs (12%) in a heterozygous form, but none of the control dogs carried two copies of an EOAD-associated haplotype. Both parents of the two affected siblings (S1 Fig) had one copy of the EOAD-associated haplotypes, indicating an autosomal recessive mode of inheritance where both the sire and dam were a carrier of the risk variant. One of the half-siblings of the affected dogs (Dog_1174) was a carrier of an EOAD-associated haplotype, whereas the other half-sibling (Dog_1106) and the dam of these two dogs (Dog_1175) did not have any EOAD-associated haplotype.

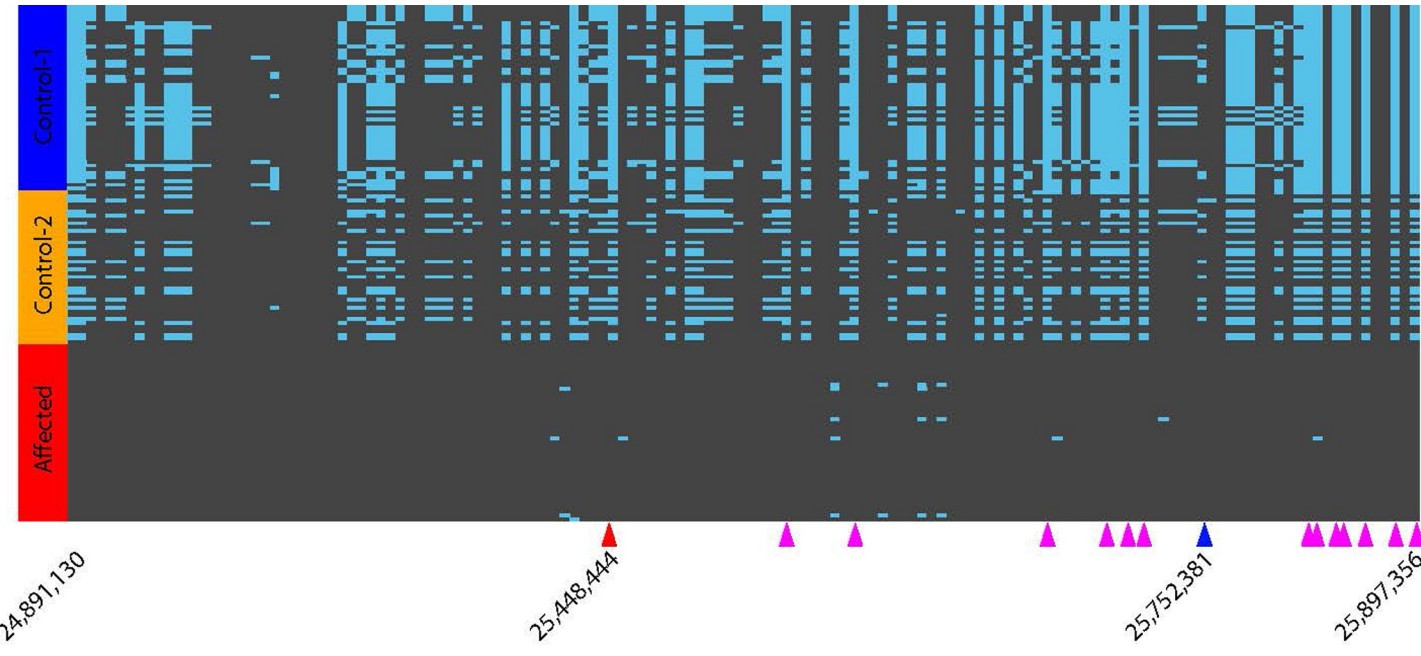

**Fig 2. Haplotypes near the marker on CFA18 that are significantly associated with Early Onset Adult Deafness (EOAD) in Rhodesian Ridgebacks.** Random subsets of control dogs were selected for visualization purpose: Control-1 dogs (blue) do not carry any of the EOAD-associated haplotypes, whereas Control-2 dogs (orange) carry one copy of the EOAD-associated haplotype (i.e., a carrier). Affected dogs are indicated in red. Rows correspond to haplotypes (two rows/individual), and columns correspond to markers. The positions of the first and last markers are 24,891,130 and 25,897,356, respectively. The most significant marker in the GWAS is indicated by a red triangle (position 25,448,444). The marker closest to the 12-bp deletion is indicated by a blue triangle (position 25,752,381). Magenta triangles indicate markers where Control-1 dogs are homozygous for a reference allele and affected dogs are homozygous for an alternate allele. These 13 informative markers, along with the most significant marker (red triangle) are used for defining a core haplotype predictive of EOAD in Rhodesian Ridgebacks. Positions of the informative markers are 25545949, 25557090, 25660957, 25696066, 25704136, 25708602, 25857995, 25861018, 25862585, 25863350, 25863894, 25883389, and 25897356. The genome coordinates in canFam3.

In order to define a core haplotype predictive of EOAD in Rhodesian Ridgebacks and identify other dog breeds that carried the same core haplotype, we chose 14 markers across a 449-kb window that were homozygous for the alternate allele in all EOAD-affected Rhodesian Ridgebacks (Fig 2). We looked for this haplotype in a sample of 110,205 purebred dogs submitted for commercial genetic testing and found 1,399 of them (1.3%) were homozygous for the haplotype. These dogs were of various breeds, with a significant number being Pembroke Welsh Corgis (1,304 dogs or 93.2%). As Pembroke Welsh Corgis were not known to exhibit EOAD, we chose this breed as a reference to screen and validate candidate causal variants linked with this core haplotype, assuming that any causal mutation found on this risk haplotype in EOAD-affected Rhodesian Ridgebacks would not be present in the Pembroke Welsh Corgis carrying the haplotype (see the **Validation of the association** section).

Whole genome sequencing of one EOAD affected dog identified 2,089 non-reference homozygous variants within the associated region. Two SNPs (CFA18:25494223T>A and CFA18:25660957T>C) were predicted to be deleterious by introducing missense mutations in *PKP3* and *LRRC56* (SIFT score = 0.02). However, these variants were probably not associated with EOAD because whole genome variants of 722 dogs and wild canid species in a NCBI database [30] showed that 63 and 65 dogs were homozygous for the non-reference alleles at these two SNP positions (allele frequencies = 0.206 and 0.162, respectively). While auditory phenotypes of these samples in the public database were unknown, they likely had normal hearing. Two deletions relative to the reference sequence were identified within the associated region, and the EOAD affected dog was homozygous for both (50-bp deletion at CFA18:25874759–25874809 and 221-bp deletion at CFA18:25885742–25885963). However, these two deletions were also found in four other Rhodesian ridgebacks reported in Plassais *et al*. [30] (S2 Table). Since the 50-bp deletion was not in linkage disequilibrium with the associated variant identified in the GWAS (CFA18:25,448,444), we concluded this deletion was unrelated to EOAD. The other 221-bp deletion was in linkage disequilibrium with the most associated GWAS marker CFA18:25,448,444. This deletion was likely accompanied by SINEC_Cf (CFA18:25885768–25885963) located at an intron of *ALKBH3*. Since this deletion was also identified in Pembroke Welsh Corgis, German Shepherd Dogs, and Labrador Retrievers (S2 Table), we did not further consider this deletion as a candidate for EOAD since this condition has not been reported in these breeds.

Among 33 protein coding genes in the EOAD-associated region based on the Ensembl gene annotation (S5 Table), *EPS8L2* (Ensembl gene ID: ENSCAFG00000025183) was of particular interest because this gene was identified as a putative causal gene for hearing loss in consanguineous families in humans [31, 32]. The Ensembl gene annotation predicted 22 exons in this gene in one of the latest dog reference genomes (UMICH_Zoey_3.1/canFam5), whereas the canFam3.1 reference genome had fewer annotated exons. The difference in the gene annotation between the two canine genome assemblies was likely due to the high GC-content, which introduced misassemblies in canFam3.1 (65.9% and 66.6% GC-content in canFam3.1 and canFam5, respectively). To identify candidate causal variants associated with EOAD, we designed primers to amplify the entire *EPS8L2* region based on the UMICH_Zoey_3.1/canFam5 reference genome (S3 Table and S3 Fig). Out of these primer pairs, 15 exonic and 11 intronic regions were partially amplified (a total length of 6.7-kb, mean fragment size of 514 bp). Sequencing these fragments identified three SNPs and two indels in 11 EOAD affected dogs. A SNP in the first intron at CFA18:25859452G>A was not perfectly associated with EOAD, whereas the other two SNPs in intron 13 (CFA18:25869159C>T) and intron16 (CFA18:25869858G>A) were perfectly or nearly perfectly associated with the EOAD status (S6 Table) (coordinates in UMICH_Zoey_3.1/canFam5). Among the two deletions that we found, the 12-bp deletion in exon 12 (CFA18:25868739–25868751) was perfectly associated

with EOAD status, where all EOAD-affected dogs were deletion homozygotes (n = 22 with available DNA, one DNA sample was not available at the time of analysis), 17 dogs with one copy of the EOAD-associated haplotypes (Fig 2) were heterozygous for the deletion, including the obligate heterozygote dogs (parents of the EOAD-affected dogs; Dog_1029 and Dog_1038; S1 Fig), and 33 dogs without the EOAD-associated haplotypes did not have the deletion (S6 Table). The other deletion was found in intron 17 (CFA18:25870908–25870912) and was not associated with the condition by having both a wildtype homozygote and a deletion homozygote in the EOAD case group (S6 Table).

## Functional prediction of the 12-bp deletion in *EPS8L2*

Ensembl Variant Effect Predictor (VEP) did not predict any strong pathogenicity in the four intronic variants in *EPS8L2*. In contrast, the 12-bp deletion likely causes an inframe shift by removing four amino acids from the EPS8L2 protein (valine, histidine, phenylalanine, and leucine: VHFL). A translated amino acid sequence of exon 12 consists of 25 amino acid residues and was highly conserved across vertebrates (S4 Fig). The four amino acid residues that were deleted in the EOAD-associated haplotypes were identical in all analyzed vertebrate species in the RefGene dataset except the Malayan flying lemur (*Galeopterus variegatus*, galVar1) and the zebrafish (*Danio rerio*, danRer11). The variant was detected as deleterious by PROVEAN (Protein Variation Effect Analyzer) [33] with the score = -29.992 (score smaller than -2.5 considered as deleterious), but MutPred-Indel [34] did not identify a sufficiently strong statistical signal to call this deletion to be pathogenic by using a Human Gene Mutation Database as a training dataset (score = 0.43471, where the score > 0.5 are considered to be pathogenic).

Folding Unit predictor (FUpred) [35] predicted that the region containing the deleted VHFL amino acids has a capability of forming an alpha helix, which was a part of a predicted protein domain D5 (S5 Fig). A predicted tertiary structure of the human and mouse EPS8L2 proteins by AlphaFold (UniProtKB: Q9H6S3 and Q99K30) showed that the VHFL amino acid residues are indeed a part of an alpha helix structure (S6 Fig).

## Validation of the association

To test the association between the 12-bp deletion at CFA18:25868739–25868751 and EOAD, we genotyped three additional EOAD-affected dogs and 12 additional control dogs by Sanger sequencing. We confirmed that all of the three EOAD-affected dogs were homozygous for the 12-bp deletion at CFA18:25868739–25868751, whereas none of the control dogs carried two copies of the deletion. Consistent with the predicted recessive mode of inheritance, the sire (dog_1186) of the two full-sibling EOAD-affected dogs (dog_1187 and dog_1188) was a carrier by having one copy of CFA18 with the deletion (S1 Table).

The core haplotype predictive of EOAD in Rhodesian Ridgebacks (Fig 2) was also common in Pembroke Welsh Corgis, but this breed is not known to exhibit EOAD. To validate the association between the 12-bp deletion and EOAD, we chose eight adult Pembroke Welsh Corgis that were homozygous for the core haplotype with no owner-reported hearing problems as controls and genotyped them for the 12-bp deletion. This analysis confirmed the absence of the deletion in these Pembroke Welsh Corgis, further supporting the association between the deletion and EOAD in Rhodesian Ridgebacks.

## Discussion

Compared with other canine hereditary deafness, hearing loss in Rhodesian Ridgebacks is epidemiologically unique by showing an early loss of hearing before reaching adulthood and not showing any noticeable association with pigmentation patterns. In this study, we identified a

12-bp deletion in *EPS8L2* by targeted Sanger sequencing and showed that the deletion was perfectly associated with EOAD in Rhodesian Ridgebacks. It should be noted that this deletion was not seen by short-read whole-genome sequencing due to an underrepresentation of this GC-rich region. The segregation pattern of the 12-bp deletion in a nuclear family is consistent with an autosomal recessive mode of inheritance, where both parents of the affected dogs carried one copy of the chromosome with the 12-bp deletion.

Because the EOAD-associated haplotypes showed extended linkage with potentially other candidate causal mutations, we could not definitively conclude a causal relationship between EOAD and the 12-bp deletion. However, the resulting inframe deletion can modify or disrupt the function of the EPS8L2 protein by shortening an alpha-helix structure. PROVEAN and MutPred-Indel predicted different significance levels of pathogenicity of this deletion likely due to the underlying difference in the prediction algorithm and reference databases. While PROVEAN is based on a pairwise sequence alignment score between the query sequence and each of the related sequences, MutPred-Indel employs a machine learning method with a neural network model that incorporates various structural and functional features, such as evolutionary conservation, protein structures, and binding capacities [33, 34]. Dahmani *et al.* [31] identified a 1-bp deletion in the same exon as a strong candidate for a childhood onset hearing loss (OMIM Allelic Variant: 6149988.0001). This mutation is predicted to cause a frameshift leading to a premature termination of translation. Given the physical proximity of this deletion found in humans and the 12-bp deletion found in this study (13-bp apart), this region likely plays a critical role in the function of the EPS8L2 protein.

*EPS8L2* is one of the epidermal growth factor receptor pathway substrate 8 (EPS) family genes and has similar protein domain organizations with other family genes, including *EPS8*, *EPS8L1*, and *EPS8L3* [36]. Experiments using *EPS8* knockout mice have shown that the EPS8 protein is involved in the growth of the hair bundle of the inner ear cells in the cochlea by controlling actin bundling and capping [37, 38]. *EPS8L2* is also expressed in the inner ear hair cells with an actin binding capability, suggesting a similar function with *EPS8* [36]. A major difference between these EPS8 family genes is that *EPS8L2* knockout mice showed progressive hearing loss accompanied by progressive deterioration of inner ear hair bundles, whereas *EPS8* knockout mice showed profound congenital deafness [37–39]. The progressive hearing loss identified in the *EPS8L2* knockout mice phenocopied childhood hearing loss in humans and EOAD observed in Rhodesian Ridgebacks, indicating that this gene is required for the maintenance of the mature hair cells in the inner ear [31, 39].

## Conclusion

We identified a novel potentially pathogenic deletion in *EPS8L2* responsible for EOAD in Rhodesian Ridgebacks by using custom canine SNP microarray genotyping and targeted sequencing. Independent identifications of pathogenic variants in *EPS8L2* in human and dog populations ascertain the key role of *EPS8L2* in the maintenance and integrity of the inner ear hair cells in mammals. Deafness is observed in various dog breeds with different clinical characteristics (e.g., congenital, environmental, age-related) and different genetic architecture. There are more than 100 genes known to be involved in non-syndromic hearing loss in humans [8], and, therefore, studying breed-specific hearing loss in dogs suggests a promising translational value toward the better understanding of the genetics of hearing loss in humans.

## Supporting information

**S1 Fig. One nuclear family with two full siblings affected with EOAD (Dog_1048 and Dog_1150).** Genotypes of the EOAD-associated 12-bp deletion at CFA18:25868739–25868751

are indicated.
(PDF)

**S2 Fig. Q-Q plots of the association with Early Onset Adult Deafness (EOAD) in Rhodesian Ridgebacks.**
(PNG)

**S3 Fig. Annotated exon and intron structure of *EPS8L2* based on Ensembl annotations in the UMICH_Zoey_3.1/canFam5 genome.** Forward and reverse primers used for PCR amplifications are indicated by red and green circles, respectively. Arrows indicate Sanger sequencing reads of PCR amplicons, where the sequences that are concordant to the reference genome are filled with red, whereas discordant sequences are shown as open symbols, representing SNPs, indels, and low quality sequence regions. The 12-bp deletion associated with EOAD is indicated (CFA18_25868739-25868751-12bp deletion in deaf dogs).
(PNG)

**S4 Fig. Exon 12 of *EPS8L2* in the human reference genome (GRCh38).** Predicted translated amino acid sequences are indicated in blue with single-letter amino acid codes. Predicted amino acid sequences in other genera/species are indicated in green (amino acids identical to human), yellow (conserved amino acid substitution), and red (non-conserved amino acid substitution). A red rectangle highlights the four amino acid deletion identified in EOAD-affected Rhodesian Ridgebacks in this study. OMIM Allelic Variant Phenotypes (614988.0001) is associated childhood hearing loss reported in Dahmani et al. (2015).
(PNG)

**S5 Fig. Prediction of protein domain structure of *EPS8L2* in dogs (UMICH_Zoey_3.1/canFam5) by FUpred.** A) Predicted secondary structure. B) A contact map showing eight potential protein domains, predicted by maximising the number of intra-domain contacts while minimizing the number of inter-domain contacts. C) FU-score plotted against domain splitting points indicated by red dotted lines, corresponding to the eight domains. D) A FU-score heat map. Predicted domain boundaries are indicated by dotted lines.
(PDF)

**S6 Fig. Predicted tertiary and secondary structure of *EPS8L2*.** A) Human EPS8L2 (UniPlot ID: Q9H6S3). B) Mouse EPS8L2 (UniPlot ID: Q99K30).
(PDF)

**S1 Table. A list of samples used in the study (both GWAS discovery study and validation) and their genotypes at the most significantly associated GWAS marker and five additional variants discovered by Sanger sequencing.**
(XLSX)

**S2 Table. Samples of whole-genome re-sequencing data used for characterizing genetic variations in the EOAD-associated region on CFA18 and their genotypes at the most significantly associated GWAS marker and two additional deletion loci.**
(XLSX)

**S3 Table. A list of primer sequences used for PCR assays and Sanger sequencing analysis.** The coordinates of the priming positions are UMICH_Zoey_3.1/canFam5.
(PDF)

**S4 Table. The number of replicated GWAS runs, where a given marker (CFA18:25,448,444, CFA24:42,714,739 and CFA26:23,554,170) is significantly associated**

**with Early Onset Adult Deafness (EOAD) in Rhodesian Ridgebacks.**
(PDF)

**S5 Table. Predicted genes by Ensembl within the EOAD-associated region on CFA18.** The gene annotation is for canFam3.1 (database version 104.31).
(PDF)

**S6 Table. Genotype frequencies of the markers in *EPS8L2*.** A) CFA18:25859452G>A in the intron 1. B) A 12-bp deletion at CFA18:25868739–25868751 in the exon 12. C) CFA18:25869159C>T in the intron 13. D) CFA18:25869858G>A in the intron 16. E) A 5-bp deletion at CFA18:25870908–25870912 in the intron 17. Genome coordinates are based on UMICH_Zoey_3.1/canFam5.
(PDF)

# Acknowledgments

We thank the Rhodesian Ridgeback Club of the United States and dedicated individual dog owners for their support and participation in this research. We thank all the Embark employees and the ProjectDOG team who made this work possible, particularly Dr. Mark Neff for providing valuable insights into the study design and inputs to the early version of the manuscript, and Calvin Leather for developing bioinformatics infrastructure.

# Author Contributions

**Conceptualization:** Takeshi Kawakami, Alison L. Ruhe, Meghan K. Jensen, Adam R. Boyko.

**Data curation:** Takeshi Kawakami, Alison L. Ruhe, Meghan K. Jensen.

**Formal analysis:** Takeshi Kawakami, Vandana Raghavan, Meghan K. Jensen, Ausra Milano, Thomas C. Nelson.

**Investigation:** Takeshi Kawakami, Vandana Raghavan, Ausra Milano.

**Methodology:** Takeshi Kawakami, Vandana Raghavan, Ausra Milano.

**Project administration:** Takeshi Kawakami, Adam R. Boyko.

**Supervision:** Takeshi Kawakami, Adam R. Boyko.

**Validation:** Vandana Raghavan, Thomas C. Nelson.

**Visualization:** Takeshi Kawakami, Vandana Raghavan, Thomas C. Nelson.

**Writing – original draft:** Takeshi Kawakami.

**Writing – review & editing:** Takeshi Kawakami, Vandana Raghavan, Alison L. Ruhe, Meghan K. Jensen, Thomas C. Nelson, Adam R. Boyko.

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
