## [Decision Letter · Decision Letter 0]

30 Dec 2021

PONE-D-21-37584Early onset adult deafness in the Rhodesian Ridgeback dog is associated with an in-frame deletion in the EPS8L2 genePLOS ONE

Dear Dr. Kawakami,

Thank you for submitting your manuscript to PLOS ONE. After careful consideration, we feel that it has merit but does not fully meet PLOS ONE’s publication criteria as it currently stands. Therefore, we invite you to submit a revised version of the manuscript that addresses the points raised during the review process.

We look forward to receiving your revised manuscript.

Kind regards,

Junwen Wang, Ph.D.

Academic Editor

PLOS ONE

Journal Requirements:

PONE-D-21-37584

3. Thank you for stating the following in the Competing Interests/Financial Disclosure * (delete as necessary) section: 

(I have read the journal’s policy and the authors of this manuscript have the following competing interests: TK, VR, ALR, MKJ, AM, TCN, and ARB are employees of Embark Veterinary, a canine DNA testing company which offers commercial testing for the variant described in this study. ARB is co-founder and part owner of Embark.)

We note that you received funding from a commercial source: (Embark)

Reviewers' comments:

Reviewer's Responses to Questions

**Comments to the Author**

1. Is the manuscript technically sound, and do the data support the conclusions?

Reviewer #1: Yes

Reviewer #2: Yes

2. Has the statistical analysis been performed appropriately and rigorously? 

Reviewer #1: Yes

Reviewer #2: Yes

3. Have the authors made all data underlying the findings in their manuscript fully available?

Reviewer #1: Yes

Reviewer #2: Yes

4. Is the manuscript presented in an intelligible fashion and written in standard English?

Reviewer #1: Yes

Reviewer #2: Yes

5. Review Comments to the Author

Reviewer #1: The data is interesting, and I agree with the conclusion made by the authors; that they could not definitely conclude a causal relationship between EOAD and the 12-bp deletion in EPS8L2 gene. I also believe that this is the most interesting finding of all variants presented.

The authors only sequenced the 12-bp deletion in eight adult Pembroke Welsh Corgis that were homozygous for the core haplotype. Would it be possible to increase that number? Where these the only ones that were homozygous for the haplotype?

In-frame deletions can be more difficult to predict in terms of pathogenicity. I think it would be interesting to further explore how the deletion affects the protein, in particularly the alpha-helix structure, and down stream signaling if possible. If the 12-bp deletion would be classified from “likely/possibly pathogenic” to “pathogenic” it could be included is breeding programs for Rhodesian Ridgebacks.

You state that the owners provided informed consent to participate in the study. Is that enough or should the study be approved by an ethics committee?

Line 291, I believe it should be “deletion”.

Figure 2 is upside down in the version I received.

Reviewer #2: Manuscript Summary

This is a well-written and relatively straightforward manuscript describing the identification of an associated variant in the EPS8L2 gene with early onset adult deafness in Rhodesian Ridgeback dogs. The paper is clear, the scientific conclusions are sound, and the manuscript overall is worthy of publication as it will make a strong contribution to the literature on hearing loss in dogs and humans. My comments are all relatively minor (except one major point, but easily addressable), however there quite a few minor points that merit addressing.

Major

1. Please indicate somewhere in the manuscript where the genotyping and WGS data is available publicly. This is mentioned in the cover page from PLoS ONE but I didn’t see it anywhere in the manuscript itself, so not sure whether/how this information will be conveyed to readers.

Minor

1. Line 24: “disorder” not “disorders” – should be singular

2. Line 55: after [8] should be a comma rather than a period

3. Line 58: “an” accurate diagnosis

4. Line 59: “the” relative contribution

5. Line 88: strike the word “Since”

6. Line 94: should be: “translational medical value for better understanding childhood…” rather than as written.

7. Lines 109-111: can you clarify how you identified these RR owners? Through the breed club? Through the Embark database?

8. Line 117: Can you clarify how/where the BAER test was done? Did you have owners bring their dogs to a veterinary neurologist and did he/she interpret the test?

9. Line 121: Awkwardly worded with “of Rhodesian Ridgebacks” – consider: “Three additional EOAD-affected and 12 control Rhodesian Ridgebacks were recruited…”

10. Line 124: Again, please clarify how the BAER tests were performed and analyzed.

11. Line 128: “are” listed

12. Lines 136-139: This really should be part of the results section rather than the methods – consider moving these sentences to the beginning of the section labeled “A novel association on chromosome 18 with EOAD”

13. Line 145: “A Wald test” rather than “Wald test”

14. Line 147: The beginning of this sentence is incomplete: “To evaluate whether marginally significant associations identified in the main GEMMA analysis,…” – either remove the word “whether” or add something like “were relevant” or “remained significant”

15. Line 150-151: I would clarify this by saying something like “We only considered regions showing consistent association….EAOD-associated region in downstream analyses.” Also – can you clarify what you mean by “consistent association”? Did it have to remain associated in 100/100 randomization experiments?

16. Line 165: Please state the VEP version

17. Line 175: should be “these breeds are” rather than “these breeds were”

18. Line 180: please add the word “region” so that it reads “to amplify regions in and around this gene”

19. Line 187: “genotype and phenotype” and should be “were used”; change “nor” to “and”

20. Line 189: I don’t think “Personal Information” should be capitalized?

21. Line 195: Can you clarify that the 23 EOAD dogs you are reporting here do NOT include the 3 “additional” EOAD dogs that you report in the methods? Same for the 162 control dogs – do these exclude the 12 additional controls (line 197)?

22. Line 202: “both had” rather than “were both”

23. Line 212: “were” rather than “are”

24. Lines 214-215: Should be “, where we repeated the GWAS analysis 100 times”

25. Line 216: please change to “of 23 case and 162 control dogs” rather than “with”

26. Lines 213-218: Can you clarify the results of these repeated experiments? What does “not detected as significantly associated regions in replicated runs” mean? In some fraction of the 100 runs? In none of them? Same with “consistently significantly associated” – was this in 100% of the replicated runs? I get the big picture here but I think some more details are warranted.

27. Lines 226-227: should be “an EOAD-associated haplotype” rather than “the EOAD-associated haplotypes.”

28. Liner 230: should be “an EOAD-associated haplotype” rather than “the”

29. Line 232: should be “was a carrier of an EOAD-associated haplotype,”

30. Line 234: should be “have any EOAD-associated haplotype.”

31. Line 243: EOAD not EAOD

32. Line 248: Kill the word “A” at the beginning of the sentence.

33. Line 253: should be “were homozygous for the non-reference alleles”

34. Line 254: consider changing “hearing conditions” to “auditory phenotypes”

35. Line 259: get rid of “the” in “the Plassais et al”

36. Lines 259-260: You can delete the sentence “The 50 bp deletion…” b/c this is repetitive from the previous sentence, and then start the next sentence: “Because the 50 bp deletion was not in LD…”

37. Line 281: should be “11 EOAD affected” not “11 of EOAD affected”

38. Line 283: “intron 13” not “the intron 13”

39. Lines 281-294: I think this paragraph could be replaced, or at least complemented by, a table showing the results for the 3 SNPs and 2 indels. You might have columns that show the position, whether SNP or indel, the location within the gene (intron vs. exon), and for both cases and controls – numbers of homozygous reference, het, and homozygous variant. It would make the point more clearly for readers. I wouldn’t require this, but just something for you to consider.

40. Line 298: I would say “likely causes” rather than “likely resulted in”

41. Line 300: “consists of” and not “consisted of”

42. Line 311: “has” not “had”

43. Line 313: “are”, not “were”

44. Line 318: I would say “three additional EOAD-affected dogs and 12 additional control dogs”

45. Line 321: delete “chromosomes with”

46. Line 331: please say “deletion and EOAD in Rhodesian Ridgebacks” for clarification.

47. Line 339: I would say “..was not seen by short-read whole genome sequencing” rather than “characterized by” as characterized implies a different type of analysis.

48. Discussion: I would consider adding a sentence or two (max) about why one of the mutation prediction programs did not classify the deletion as being pathogenic (MutPred) but the other one did (PROVEAN).

6. PLOS authors have the option to publish the peer review history of their article (what does this mean?). If published, this will include your full peer review and any attached files.

Reviewer #1: No

Reviewer #2: No

---

## [Author Response · Author response to Decision Letter 0]

20 Jan 2022

Reviewers' comments:

AU: We are happy that both reviewers found the paper interesting and exciting. We have carefully revised the paper to address the reviewer’s concerns. Our responses are marked as “AU:” 

Reviewer #1: The data is interesting, and I agree with the conclusion made by the authors; that they could not definitely conclude a causal relationship between EOAD and the 12-bp deletion in EPS8L2 gene. I also believe that this is the most interesting finding of all variants presented.

The authors only sequenced the 12-bp deletion in eight adult Pembroke Welsh Corgis that were homozygous for the core haplotype. Would it be possible to increase that number? Where these the only ones that were homozygous for the haplotype?

AU: We do not think that additional genotyping of the deletion in Pembroke Welsh Corgis will change our conclusion. We demonstrated the perfect association between the EOAD-associated haplotypes and the deletion in 22 homozygous and 17 heterozygous Rhodesian Ridgebacks. This means that all 61 chromosomes with the EOAD-associated haplotypes that we examined carried the deletion in this breed. In contrast, all 16 chromosomes in the eight Pembroke Welsh Corgis that we tested did not have the deletion. This should provide sufficiently strong evidence for the breed-specific association between the haplotype and the deletion even if the core haplotype based on the array markers is identical in these two breeds

In-frame deletions can be more difficult to predict in terms of pathogenicity. I think it would be interesting to further explore how the deletion affects the protein, in particularly the alpha-helix structure, and down stream signaling if possible. If the 12-bp deletion would be classified from “likely/possibly pathogenic” to “pathogenic” it could be included is breeding programs for Rhodesian Ridgebacks.

AU: We fully agree with the reviewer. Experimental evidence for the structural change of the protein induced by the 12-bp deletion can advance our understanding of its pathogenicity. Unfortunately, empirical analysis in structural biology, such as X-ray crystallography and 3D electron microscopy, is out of the scope of this study. As the reviewer pointed out, direct genotyping of the 12-bp deletion should provide valuable information for many Rhodesian Ridgeback breeders to make an informed decision in their breeding programs.

You state that the owners provided informed consent to participate in the study. Is that enough or should the study be approved by an ethics committee?

AU: Our study fully complies with the journal guideline (https://journals.plos.org/plosone/s/submission-guidelines). All samples were collected by buccal swabbing by dog owners, and phenotyping by BAER testing was conducted by veterinary clinics. As stated in lines 214-216, these genotyping and phenotyping methods are non-invasive (no anesthesia, no sedation, no prolonged physical and mental stresses were applied), and dogs were never handled directly by researchers.

Line 291, I believe it should be “deletion”.

AU: Corrected

Figure 2 is upside down in the version I received.

AU: We apologize for the inconvenience. It must have something to do with the manuscript uploading process. We will certainly double-check the final proofs to ensure it is correct in the published version. 

Reviewer #2: Manuscript Summary

This is a well-written and relatively straightforward manuscript describing the identification of an associated variant in the EPS8L2 gene with early onset adult deafness in Rhodesian Ridgeback dogs. The paper is clear, the scientific conclusions are sound, and the manuscript overall is worthy of publication as it will make a strong contribution to the literature on hearing loss in dogs and humans. My comments are all relatively minor (except one major point, but easily addressable), however there quite a few minor points that merit addressing.

Major

1. Please indicate somewhere in the manuscript where the genotyping and WGS data is available publicly. This is mentioned in the cover page from PLoS ONE but I didn’t see it anywhere in the manuscript itself, so not sure whether/how this information will be conveyed to readers.

AU: We stated that all data generated in this study are freely available in the Data Availability Statement section. We revised these statements for further clarification (Line 628-633). 

“All data necessary to replicate our results, including the array genotype data, are freely available on Dryad with the provided DOI (doi:10.5061/dryad.pg4f4qrqt). Raw sequence reads of the whole genome sequencing analysis are available via the Short Read Archive (ncbi.nlm.nih.gov/sra; Bioproject number: PRJNA780814). Sanger sequence results are available in Genbank (GenBank accession: OL652584-OL652592).”

Minor

1. Line 24: “disorder” not “disorders” – should be singular

AU: Corrected

2. Line 55: after [8] should be a comma rather than a period

AU: Corrected

3. Line 58: “an” accurate diagnosis

AU: Corrected

4. Line 59: “the” relative contribution

AU: Corrected

5. Line 88: strike the word “Since”

AU: Corrected

6. Line 94: should be: “translational medical value for better understanding childhood…” rather than as written.

AU: Corrected

7. Lines 109-111: can you clarify how you identified these RR owners? Through the breed club? Through the Embark database?

AU: The sentence was revised:

In order to recruit EOAD-affected and normal hearing dogs, we conducted an online survey in 2020 to private dog owners of Rhodesian Ridgebacks who were registered in the Embark customer database and agreed to participate in scientific research.

8. Line 117: Can you clarify how/where the BAER test was done? Did you have owners bring their dogs to a veterinary neurologist and did he/she interpret the test?

AU: We added a sentence:

BAER testing was coordinated by ProjectDog through individual clinics and in-home visits.

9. Line 121: Awkwardly worded with “of Rhodesian Ridgebacks” – consider: “Three additional EOAD-affected and 12 control Rhodesian Ridgebacks were recruited…”

AU: Revised as suggested

10. Line 124: Again, please clarify how the BAER tests were performed and analyzed.

AU: The sentence was revised:

These three cases were confirmed to be EOAD by observing early loss of hearing by the owners, followed by BAER testing, which was conducted at individual clinics

11. Line 128: “are” listed

AU: Corrected

12. Lines 136-139: This really should be part of the results section rather than the methods – consider moving these sentences to the beginning of the section labeled “A novel association on chromosome 18 with EOAD”

AU: Two sentences in lines 134-139 were moved to the result subsection “A novel association on chromosome 18 with EOAD” 

13. Line 145: “A Wald test” rather than “Wald test”

AU: Corrected

14. Line 147: The beginning of this sentence is incomplete: “To evaluate whether marginally significant associations identified in the main GEMMA analysis,…” – either remove the word “whether” or add something like “were relevant” or “remained significant”

AU: Corrected

15. Line 150-151: I would clarify this by saying something like “We only considered regions showing consistent association….EAOD-associated region in downstream analyses.” Also – can you clarify what you mean by “consistent association”? Did it have to remain associated in 100/100 randomization experiments?

AU: Corrected. “Consistent association” was clarified.

16. Line 165: Please state the VEP version

AU: The version number was added

17. Line 175: should be “these breeds are” rather than “these breeds were”

AU: Corrected

18. Line 180: please add the word “region” so that it reads “to amplify regions in and around this gene”

AU: Added

19. Line 187: “genotype and phenotype” and should be “were used”; change “nor” to “and”

AU: Corrected

20. Line 189: I don’t think “Personal Information” should be capitalized?

AU: Corrected

21. Line 195: Can you clarify that the 23 EOAD dogs you are reporting here do NOT include the 3 “additional” EOAD dogs that you report in the methods? Same for the 162 control dogs – do these exclude the 12 additional controls (line 197)?

AU: We added the following sentences/phrases: 

“This dataset was used for GWAS to identify genomic regions and variants associated with EOAD (hereafter referred to as a discovery dataset).” (Line 129-)

“These samples were used in the downstream analyses to identify EOAD-associated regions by GWAS and screen candidate variants by haplotype analysis.” (Line 236-)

22. Line 202: “both had” rather than “were both”

AU: Corrected

23. Line 212: “were” rather than “are”

AU: Corrected

24. Lines 214-215: Should be “, where we repeated the GWAS analysis 100 times”

AU: Corrected

25. Line 216: please change to “of 23 case and 162 control dogs” rather than “with”

AU: Corrected

26. Lines 213-218: Can you clarify the results of these repeated experiments? What does “not detected as significantly associated regions in replicated runs” mean? In some fraction of the 100 runs? In none of them? Same with “consistently significantly associated” – was this in 100% of the replicated runs? I get the big picture here but I think some more details are warranted.

AU: To clarify the sentence, we added the following sentence (Line 259-)

“CFA24:42,714,739 was not significantly associated with EOAD in any of the replicated runs, and CFA26:23,554,170 was significantly associated with EOAD in 14 of the 100 replicated runs.”

27. Lines 226-227: should be “an EOAD-associated haplotype” rather than “the EOAD-associated haplotypes.”

AU: Corrected

28. Liner 230: should be “an EOAD-associated haplotype” rather than “the”

AU: Corrected

29. Line 232: should be “was a carrier of an EOAD-associated haplotype,”

AU: Corrected

30. Line 234: should be “have any EOAD-associated haplotype.”

AU: Corrected

31. Line 243: EOAD not EAOD

AU: Corrected

32. Line 248: Kill the word “A” at the beginning of the sentence.

AU: Corrected

33. Line 253: should be “were homozygous for the non-reference alleles”

AU: Corrected

34. Line 254: consider changing “hearing conditions” to “auditory phenotypes”

AU: Corrected

35. Line 259: get rid of “the” in “the Plassais et al”

AU: Corrected

36. Lines 259-260: You can delete the sentence “The 50 bp deletion…” b/c this is repetitive from the previous sentence, and then start the next sentence: “Because the 50 bp deletion was not in LD…”

AU: Corrected

37. Line 281: should be “11 EOAD affected” not “11 of EOAD affected”

AU: Corrected

38. Line 283: “intron 13” not “the intron 13”

AU: Corrected

39. Lines 281-294: I think this paragraph could be replaced, or at least complemented by, a table showing the results for the 3 SNPs and 2 indels. You might have columns that show the position, whether SNP or indel, the location within the gene (intron vs. exon), and for both cases and controls – numbers of homozygous reference, het, and homozygous variant. It would make the point more clearly for readers. I wouldn’t require this, but just something for you to consider.

AU: Thank you for picking this up. It is our mistake that we failed to refer to Table S6 in the body text. I suspect that we accidentally dropped the reference during the review process. We referenced the table in an appropriate place (Lines 346, 353, 356)

40. Line 298: I would say “likely causes” rather than “likely resulted in”

AU: Corrected

41. Line 300: “consists of” and not “consisted of”

AU: Corrected

42. Line 311: “has” not “had”

AU: Corrected

43. Line 313: “are”, not “were”

AU: Corrected

44. Line 318: I would say “three additional EOAD-affected dogs and 12 additional control dogs”

AU: Corrected

45. Line 321: delete “chromosomes with”

AU: Corrected

46. Line 331: please say “deletion and EOAD in Rhodesian Ridgebacks” for clarification.

AU: Corrected

47. Line 339: I would say “..was not seen by short-read whole genome sequencing” rather than “characterized by” as characterized implies a different type of analysis.

AU: Corrected

48. Discussion: I would consider adding a sentence or two (max) about why one of the mutation prediction programs did not classify the deletion as being pathogenic (MutPred) but the other one did (PROVEAN).

AU: We added a sentence in Line 420-427

“PROVEAN and MutPred-Indel predicted different significance levels of pathogenicity of this deletion likely due to the underlying difference in the prediction algorithm and reference databases. While PROVEAN is based on a pairwise sequence alignment score between the query sequence and each of the related sequences, MutPred-Indel employs a machine learning method with a neural network model that incorporates various structural and functional features, such as evolutionary conservation, protein structures, and binding capacities [33,34].”

---

## [Decision Letter · Decision Letter 1]

9 Feb 2022

Early onset adult deafness in the Rhodesian Ridgeback dog is associated with an in-frame deletion in the EPS8L2 gene

PONE-D-21-37584R1

Dear Dr. Kawakami,

We’re pleased to inform you that your manuscript has been judged scientifically suitable for publication and will be formally accepted for publication once it meets all outstanding technical requirements.

Kind regards,

Regie Lyn Pastor Santos-Cortez, M.D., Ph.D.

Academic Editor

PLOS ONE

Additional Editor Comments (optional):

Reviewers' comments:

Reviewer's Responses to Questions

**Comments to the Author**

1. If the authors have adequately addressed your comments raised in a previous round of review and you feel that this manuscript is now acceptable for publication, you may indicate that here to bypass the “Comments to the Author” section, enter your conflict of interest statement in the “Confidential to Editor” section, and submit your "Accept" recommendation.

Reviewer #1: All comments have been addressed

Reviewer #2: All comments have been addressed

2. Is the manuscript technically sound, and do the data support the conclusions?

Reviewer #1: Yes

Reviewer #2: Yes

3. Has the statistical analysis been performed appropriately and rigorously? 

Reviewer #1: Yes

Reviewer #2: Yes

4. Have the authors made all data underlying the findings in their manuscript fully available?

Reviewer #1: Yes

Reviewer #2: Yes

5. Is the manuscript presented in an intelligible fashion and written in standard English?

Reviewer #1: Yes

Reviewer #2: Yes

6. Review Comments to the Author

Reviewer #1: (No Response)

Reviewer #2: Thank you for addressing my concerns and for preparing a well-written manuscript. This should be well-received by the community.

7. PLOS authors have the option to publish the peer review history of their article (what does this mean?). If published, this will include your full peer review and any attached files.

Reviewer #1: No

Reviewer #2: No

---

## [Editor Report · Acceptance letter]

28 Mar 2022

PONE-D-21-37584R1 

Early onset adult deafness in the Rhodesian Ridgeback dog is associated with an in-frame deletion in the *EPS8L2* gene 

Dear Dr. Kawakami:

I'm pleased to inform you that your manuscript has been deemed suitable for publication in PLOS ONE. Congratulations! Your manuscript is now with our production department. 

Kind regards, 

on behalf of

Dr. Regie Lyn Pastor Santos-Cortez 

Academic Editor

PLOS ONE